# Healthcare providers experiences with shared medical appointments for heart failure

Vanessa Marshall[1,2,3‡], Jeri Jewett-Tennant[4‡], Jeneen Shell-Boyd[1‡], Lauren Stevenson[1‡], Rene Hearns[1‡], Julie Gee[1], Kimberley Schaub[1], Sharon LaForest[1], Tracey H. Taveira[5,6,7], Lisa Cohen[5,6], Melanie Parent[5], Sandesh Dev[8☯], Amy Barrette[5], Karen Oliver[5], Wen-Chih Wu[5,6,7,8☯], Sherry L. Ball[1☯‡]*

1 Research, VA Northeast Ohio Healthcare System, Cleveland, Ohio, United States of America, 2 University Hospitals Cleveland Medical Center, Cleveland, Ohio, United States of America, 3 Case Comprehensive Cancer Center, Case Western Reserve University, Cleveland, Ohio, United States of America, 4 Health System Education/Opioid Data 2 Action, The Center for Health Affairs, Cleveland, Ohio, United States of America, 5 Medicine, Providence VA Medical Center, Providence, Rhode Island, United States of America, 6 College of Pharmacy, University of Rhode Island, Kingston, Rhode Island, United States of America, 7 The Warren Alpert School of Medicine, Brown University, Providence, Rhode Island, United States of American, 8 Medicine, Southern Arizona VA Health Care System, Tucson, Arizona, United States of America

☯ These authors contributed equally to this work.
‡ VM, JJT, JSB, LS, RH and SLB also contributed equally to this work.
* sherry.ball@va.gov

**Data Availability Statement:** Data cannot be shared publicly because of the VHA protection rules and the approved Data Management Action Plan. A Limited de-identified Data Set will be

## Abstract

Shared medical appointments (SMAs) offer a means for providing knowledge and skills needed for chronic disease management to patients. However, SMAs require a time and attention investment from health care providers, who must understand the goals and potential benefits of SMAs from the perspective of patients and providers. To better understand how to gain provider engagement and inform future SMA implementation, qualitative inquiry of provider experience based on a knowledge-attitude-practice model was explored. Semi-structured interviews were conducted with 24 health care providers leading SMAs for heart failure at three Veterans Administration Medical Centers. Rapid matrix analysis process techniques including team-based qualitative inquiry followed by stakeholder validation was employed. The interview guide followed a knowledge-attitude-practice model with a priori domains of knowledge of SMA structure and content (understanding of how SMAs were structured), SMA attitude/beliefs (general expectations about SMA use), attitudes regarding how leading SMAs affected patients, and providers. Data regarding the patient referral process (organizational processes for referring patients to SMAs) and suggested improvements were collected to further inform the development of SMA implementation best practices. Providers from all three sites reported similar knowledge, attitude and beliefs of SMAs. In general, providers reported that the multi-disciplinary structure of SMAs was an effective strategy towards improving clinical outcomes for patients. Emergent themes regarding experiences with SMAs included improved self-efficacy gained from real-time collaboration with providers from multiple disciplines, perceived decrease in patient re-hospitalizations, and promotion of self-management skills for patients with HF. Most providers reported that the SMA-setting facilitated patient learning by providing opportunities for the sharing of experiences and knowledge. This was associated with the perception of

created and shared pursuant to a Data Use Agreement (DUA) appropriately limiting use of the dataset and prohibiting the recipient from identifying or re-identifying (or taking steps to identify or re-identify) any individual whose data are included in the dataset. Contact the VA Northeast Ohio Healthcare System Research Office at 216-791-3800 x64660 for further information. Email requests for data can be made to Holly Henry, Administrative Officer for Research, at holly.henry@va.gov.

**Funding:** This work was supported by the VA HSRD https://www.hsrd.research.va.gov/ Merit Review Grant IIR 14-293 to W-CW. The funders had no role in study design, data collection and analysis, decision to publish, or preparation of the manuscript.

**Competing interests:** The authors have declared that no competing interests exist.

increased comradery and support among patients. Future research is needed to test suggested improvements and to develop best practices for training additional sites to implement HF SMA.

## Introduction

Heart failure (HF) is a major public health problem affecting over 6 million people in the United States with an estimated cost greater than $30 billion each year [1]. Approximately 20% of all patients hospitalized for HF within the Veterans Health Administration (VHA) are rehospitalized within 30 days, illustrating the burden on patients and the healthcare system [2]. Effective self-management strategies such as a healthy diet, medication adherence, and recognizing when to seek treatment can improve quality of life and reduce hospitalizations for people who are diagnosed with HF, especially when unhealthy behaviors and risk factors are addressed [3]. Yet, HF remains a complex chronic disease where comprehensive patient-centered care that includes supportive and medical care services including dietary, social, pharmacy and rehabilitative services are provided to the patient in coordination with primary and specialty care is difficult to achieve [4].

In a shared medical appointment (SMA) multiple patients with a common chronic illness are seen together by at least two healthcare providers often with expertise in nursing, psychology, medication management, and nutrition in single appointment to provide a forum for patients to learn about self-care from providers and through interactive support from peers [5–10]. Shared medical appointments offer a potential means of reducing rehospitalizations by providing education and support implementing self-management strategies and follow-up care. Session topics are often adapted to meet the current needs of the patient participants.

While this care model has been shown to improve patient experience and clinical outcomes in a variety of settings, the refinement of this model continues to be informed by qualitative, quantitative, and mixed methods studies. Participation in SMAs has been shown to improve patients' self-management skills [9]. However, few studies have focused on how this care model affects provider experience.

While use of SMAs is growing, further spread is expected to improve outcomes. Designing SMA programs that enhance provider and patient experience could facilitate the spread of SMAs by promoting provider buy-in and improving program sustainability. Most studies of SMAs focus on patient outcomes with few reporting on effects on healthcare providers with those that do reporting mixed results [5–10]. A better understanding of SMAs from the perspective of healthcare providers who refer patients to or lead SMAs could inform better implementation and delivery of the SMA model to ultimately improve better self-management of HF for patients.

A goal of this study was to inform sustainability and scalability of SMAs by exploring the experience of health care providers leading and referring patients to SMAs. This study used a knowledge, attitudes and practice model to explore the experience of health care providers who lead or referred patients to SMAs at three VHA sites. Data were analyzed using a qualitative rapid matrix analysis in order to quickly provide actionable feedback to the providers/respondents [11, 12]. How providers' SMA-experience can inform future SMA implementation is the focus of this paper.

## Materials and methods

This qualitative study was a portion of a VHA three-site randomized control trial (RCT) to study outcomes associated with the use of SMAs following hospitalization for HF [13]. The

qualitative work included interviews with health care providers involved in the conducting or referring patients to SMAs, the intervention arm of the RCT, and was approved by the Louis Stokes Cleveland Department of Veterans Affairs Medical Center's (LSCDVAMC) Institutional Review Board (IRB).

## Setting

All three sites involved had comparable ongoing HF SMA programs led by multi-disciplinary teams including dieticians, psychologists, physicians, nurses, clinical pharmacists, physician assistants, and nurse practitioners. All programs conducted 90-minute group medical appointments that included an assessment of patient needs, a planned topical self-management education session, and a patient-initiated disease-management discussion. Sessions included individualized attention from a healthcare provider and one site included skills training. Patients were referred to HF SMAs by either a physician, nurse, clinical pharmacist, physician assistant or nurse practitioner that were adapted to include consistent patient education. Further details of the HF SMA self-care topics and appointment protocol have been previously published [13].

## Respondents

HF SMA team leads from each site supplied names of providers leading and/or referring patients to the HF SMAs. All 33 individuals included on the list of provided names were invited by email to participate in an in-person or telephone interview regarding their experiences with HF SMAs. Of the contacted providers, 73% (33/24) responded and were interviewed. Twenty-four interviews were completed either in-person (n = 3) or by telephone (n = 21) with physicians (n = 8), pharmacists (n = 4), dieticians (n = 2), clinical psychologists (n = 4), nurse practitioners (n = 2), and registered nurses (n = 4) across three VAMC sites. Participation in this study was voluntary and confidential. Table 1 provides the distribution of respondents among sites and gender.

## Data collection

Consents were mailed by USPS to respondents and included language agreeing to both participation in the interview and allowing the interview to be recorded. After study staff received the signed consent by USPS, interviews were scheduled either in-person or by telephone. A semi-structured interview guide with grounded probes (S1 Appendix Interview Guide) was developed based on a knowledge-attitude-practice (KAP) framework which proposes that knowledge and attitude about a intervention can inform the practice of a public health intervention [14–16]. The KAP was used as a framework to identify domains important to clinician participation in SMAs rather than a predictive model. Neutral domains or data categories without positive or negative assignments were identified for each interview question. Each semi-structured interview was conducted by two of three individuals who identified as female and were experienced in semi-structured interviewing and qualitative analysis (JJ, SB, VM). Interviewees

**Table 1. Respondents.**

|  | Site A | Site B | Site C | Total |
|---|---|---|---|---|
| Referred | 19 | 8 | 6 | 33 |
| Interviewed | 10 | 8 | 6 | 24 |
| Total Referred Male/Female | 5/14 | 4/4 | 2/4 | 11/22 |
| Total Interviewed Male/Female | 2/8 | 4/4 | 2/4 | 8/16 |

included healthcare team members who led or referred patients to HF SMAs. Each interview was completed by two trained interviewers, one serving as lead interviewer and the other serving as notetaker. All respondents consented to both the interview and allowed the interview to be audio recorded. Transcripts were reviewed with audio to check transcript accuracy. A debrief form was completed immediately following the interview to record the interviewer's and notetaker's initial feedback (S2 Appendix Agenda for Debriefing Post Interview Meetings). The three-person interview team met to discuss the interview, consider and discuss potential personal biases, and build consensus on the content of the debrief form. Although all interviewers were from one of the three study sites, the interviewers had no prior relationship with the respondents at the time of the interview but communicated with the clinical teams to feedback aggregate data and obtain validation of results.

## Data analysis

Qualitative data analysis was completed in additive steps. Debrief notes and transcripts were analyzed using a team-based qualitative rapid assessment process [17] shown to expeditiously yield high quality review of data in a timely manner allowing for timely feedback to the SMA teams [18]. As an initial step, each interview question was assigned a neutral domain as listed in Table 2 and a summary template was created based on these domains: knowledge of SMA structure, SMA attitude, effect on patients, suggested improvements, experience of providers, patient referrals.

Each qualitative analyst applied the summary template to the same three transcripts and then met to reach consensus on operational definitions of each summary template domain. Once consensus was reached each transcript was assigned to an individual team member for summary completion. Team members individually reviewed debrief forms for each interview before completing corresponding summaries. Representative quotes were noted during the rapid analysis process on the summary templates. Upon completion and consensus from all analysts, summaries were compiled into a single matrix organized by site and provider type for comparison. Analysts met regularly until consensus on the entire matrix was reached.

These consensus meetings resulted in a final matrix that was condensed to five domains as all items first coded as SMA attitudes were duplicative of other domains. These five domains were arranged in columns with data from each interview summarized by row allowing for a quick assessment of domains across sites. For each site, data summaries including descriptions of SMAs were compiled and shared with the respective respondents.

## Results

Data aggregated by each of the three sites are summarized in Table 3 illustrating the many similarities and the few differences between sites. Respondents found HF SMAs to be an effective care model that offer benefits for providers and patients. Most respondents leading HF SMAs in this study found their participation as beneficial for themselves and Veterans. Respondents

**Table 2. Matrix domains and operational definitions.**

| Domains | Definition |
|---|---|
| Knowledge: SMA structure and content | Respondents' understanding of how SMA were structured at their site impact of SMAs |
| Knowledge: Patient referrals | Organizational processes for referring patients to SMAs |
| Attitude: Effect on patients | Influence of SMAs specific to patients |
| Attitude: Experience of providers | How leading SMA affected provider |
| Practice: Suggested improvements | Respondents' suggestions for improvements |

**Table 3. Matrix for all sites.**

| Summaries | Site A | Site B | Site C |
|---|---|---|---|
| SMA structure & content | • Content adjusted based on group interest and need | • Structure is adaptable. | • Multi-disciplinary |
| | • Interactive | • Set curriculum | • Individual and group attention |
| | • Multidisciplinary | • Taught like classes, but opportunity for interaction | • 4 sessions, 1.5 hour |
| | • Resources provided to patients | • Multidisciplinary; | • Need 4–5 patients to be effective |
| | • Individualized care | • Informational handouts distributed | |
| | • Group setting, 8–10 patients | • Patients get brief exam. | |
| | • Patients learn skills. | • Four different sessions; 1/week | |
| | • 2-hour comprehensive & less fragmented appointment | • Cardiologist oversight | |
| | • Provides education for medications & diet | • Critical for pharmacist to lead | |
| | • Skills of leader are critical. | | |
| Effect on patients | • Benefit from sharing knowledge & experiences. | • learning from each other | • Patient-to-patient sharing/support |
| | • Learn self-management | • Develop community | • Most successful SMA patient has support at home and is not a substance abuser or mentally ill/demented |
| | • Develop comradery with other patients. | • Receiving HF education helps with lifestyle change and self-management. | • Good for newly diagnosed patients |
| | • Convenient | • Helps with medication adherence | • Efficient use of provider/patient time |
| | • Holistic care | • Helps with emotional response to HF | |
| | • Patients become proactive. | • Validates patient's experience | |
| | • Better care continuity | • Some patients don't like groups. | |
| | • Perceived decrease in hospital and/or emergency department visits | • Some patients need more individualized attention. | |
| | • Earlier appointments | • Travel can be a barrier. | |
| | | • Easy access to providers | |
| | | • Not for very ill patients | |
| | | • Study will see how well it works | |
| Suggested improvements | • Increase number of SMAs | • Increase number of sessions | • Add exercise component |
| | • Add more providers such as nurse practitioners | • Add an exercise physiologist | • Refresher sessions would be helpful to patients |
| | • More 'new' resources: physical therapist/social worker/ | • Offer SMAs at outpatient clinics | • Need good communication between providers |
| | exercise physiologist | • Offer a support group | • Better for success when SMAs are endorsed by VA administration and/or by cardiology department heads |
| | • Offer more SMAs at outpatient clinics (especially rural) | • Add an advanced class | |
| | Longer duration | | |
| | • Encourage caregiver of patients with cognitive issues to attend | | |
| | • Add cooking class | | |
| Experience of providers | • Inter-disciplinary knowledge sharing | • Providers learn from each other. | • Able to treat patients more holistically through SMAs |
| | • Efficient sessions | • Address issues providers don't have time for | • A lot of work for provider but great for patients |
| | • Better job satisfaction | • Not helpful to providers | • Saves providers time |
| | • Learn from patient to patient interactions | • Reduces redundancy for providers | • A lot of work |
| | • Able to be more holistic with care | | |
| | • Provides opportunity for more communication between SMA providers and primary care provider (PCP)s | | |

*(Continued)*

**Table 3.** (Continued)

| Patient referrals | • Patients with new onset, existing, acute chronic symptoms or based on chart review are referred. | • HF inpatients are referred by nurse | • HF nurse approaches inpatients for immediate consent & scheduling |
|---|---|---|---|
| | • Nurse Practitioner (NP) or PCP refers. | • All HF hospitalized patients referred | • Recruitment is an issue if cardiology department isn't on-board |
| | • HF NP is SMA gatekeeper | • All HF Consults referred to SMA | • No direct consult for SMAs |
| | • Don't refer patients who don't like groups or have severe behavioral or violence issues. | • PCPs and pharmacist refer | |
| | • More direction needed for referrals | • PCPs can refer patients to specific SMA session | |
| | • Distribute more information to patient on SMA pros and cons prior to visit | • SMA provider is added as signer | |
| | | • number of referrals up since new chief | |
| | | • Streamline referral process | |

felt SMAs were beneficial due to knowledge gained from other providers suggesting potential improvements in collaborative and coordinated patient care. However, from the provider perspective, challenges were noted such as labor-intensive preparation and setup. On the other hand, this model was noted to offer potential improvements in clinical outcomes for patients.

The following provides the results organized by the neutral domains with representative quotations from respondents.

## SMA structure and content

SMAs from all sites shared commonalities in SMA structure and content. Respondents from each of the three sites reported that SMAs were conducted as consecutive multi-disciplinary interactive weekly group sessions with each session lasting approximately 90 minutes in duration. While each session included pre-determined instruction on issues such as diet and medication, each session was also adapted to correspond with the needs and requests of the patient attendees. Respondents from all sites described how a portion of each session was devoted to providing patients individualized one-on-one medical care. One site included skills training and that was the only noted content and structure related difference between the three sites.

Each respondent described the multidisciplinary nature of the SMAs. Respondents from all three sites commented on how both patients and providers learn from SMA leaders with a range of skills and knowledge.

"I think they're [SMA] excellent. I think they're a great way for people to have access to all the different team members without having to come in for many different appointments and it's also good for the team [SMA] they learn from each other." *Site B*

"I think they're really helpful. They meet a need that no one else had been able to do in the sense of these are patients that kind of need a little bit of everything in order to support them through what I think is a far harder process than I think people give credit for in terms of managing heart failure. I think it's efficient. . ." *Site A*

## Effect on patients

Most respondents emphasized how the SMA setting efficiently provided patients the opportunity to share experiences and gain knowledge from fellow participants. Respondents noted

that comradery and support were fostered as patients became familiar with one another as they attended multiple consecutive sessions together. Respondents felt feedback from fellow SMA attendees helped validate patients' emotional experiences with chronic disease management in addition to promoting better self-management for patients.

"when you see the sharing and empowering that is, goes on between patients by setting their worries at ease and that's something extremely valuable and maybe not be addressed in the individual setting because patients often don't know or not aware of other patients feel the same way about their illness." *Site B*

"I think it would be very helpful for them [the Veterans], mainly because they don't feel—there's all these underlying processes that occur in a group, and one is that they don't feel alone. That a medical condition is normalized. There's a lot of support that occurs. . ." *Site C*

Some respondents cautioned that SMAs weren't for all patients such as those who were uncomfortable in a group setting or were at an advanced stage of their chronic illness.

"I wouldn't refer psychiatric patients or those who really don't like group appointments, very private people. Newly diagnosed people would be good for SMAs." *Site A*

"I think from previous experiences patients who are very sick do not do well here because they are recently discharged from the hospital . . . they do not do well because first of all they are not that interactive with the group and they probably want all of the attention to self because they do not feel right and therefore I don't think the shared medical appointment will work if the patient is not really eager to learn or interact because they are so bogged down by their illness." *Site B*

## Suggested improvements

All sites provided suggestions to increase access to SMAs for Veterans who currently participate and to make it easier for more Veterans to participate. Some suggested adding more sessions in general while others specifically requested the addition of a class with an exercise focus. The inclusion of a refresher support group and/or follow-up sessions were also suggested. Sites felt that providers from other disciplines could participate such as an exercise physiologist. Communication between providers and support from their local cardiology chief or national leadership were noted as important to continued promotion of the service. One site stressed the importance of encouraging attendance by patients' caregivers.

"Maybe access. Being offered more in the rural areas. Some of these Veterans come from a long distance." *Site A*

"It would be nice if it can be offered on more days of the week and at different times because I know that transportation can be potentially an issue." *Site B*

"better ways to communicate about the program, getting the word out there" *Site C*

## Experience of providers

Most session leaders reported gaining knowledge by observing providers from other disciplines deliver care and gaining a better understanding of patients by observation of patient-

patient interactions during the SMAs. These respondents added that SMAs could be an efficient use of the providers' time and could contribute to improved job satisfaction.

"Excellent, and they're helpful to me in a very similar way as they are to the patients. You know, where you're involved in a group you learn all the different ways all the providers contribute, so just being involved in that I've learned all about medication for heart failure, and diet for heart failure, and symptom management for heart failure than I would have being just a sole psychologist on a team." *Site A*

". . . what we found out was that not only were we as the practitioners experts, but the patients in the room are also experts, and they were able to collaborate and share their experiences and their ideas with each other, which in my experience, has been way more impactful than just us telling them again and again what they should do or need to do, so that has really been insightful for myself, being involved with the shared medical appointments." *Site C*

However, two respondents, each from different sites, felt that sessions required lot of work including scheduling.

"So, it's possible that the scheduling could be more streamlined, or user-friendly." *Site B*

"I think it's a ton of behind-the-scenes work for the staff who are putting everything together." *Site C*

## Patient referrals

Nurses played a key role in the referral process at all sites. Overall, all sites expressed a need for improvement in the referral processes. Identifying specific guidelines for the type of patient best suited for SMAs and developing informational materials specifically directed toward potential patients was recommended.

"I am thinking that right now the referral process is not easy. Basically, we need to copy the people who are running the shared medical appointments, so they know patients that want to get a shared medical appointment. So, in the future probably something that is like a consult is needed." *Site B*

## Discussion

This qualitative study expands upon other studies examining providers' experiences with SMAs [19, 20] and focuses on providers' knowledge of structure, attitudes about effectiveness, and ideas on how to improve the practice of SMAs. The findings support other work highlighting the importance of a collaborative multi-disciplinary team to a successful SMA program [5, 21–24] and more recently discussed by Thompon-Lastad and Gardiner [25]. These data highlight how the SMA environment facilitates provider learning about multidisciplinary patient care through direct observation of providers during the SMA sessions [24]. Additionally, this study illustrates how this environment can support better care by enhancing opportunities for communication and care coordination among providers similar to other studies [19–21]. While in this study providers differed in their opinion on whether SMAs are an efficient use of providers' time all respondents noted the unique knowledge gained by observations of patient-patient and patient-provider interactions.

Having good communication between team members from multiple disciplines was noted in this and other studies [20, 21] to be elemental to a successful SMA program. This setting may promote communication among participants [22] the SMA team members and with primary care providers. In addition to the report of better provider-to-provider communication, this study supports other work suggesting that the inclusion of a designated team member skilled at facilitating group discussions could promote focused group sharing and learning among patients [26].

Opportunities to improve the SMA have focused on patient education [27] experience [20] with one study noting that some patients found the information provided to be overwhelming [7]. In contrast, data presented here suggest a need to expand awareness SMAs to both patients and providers and to increase patient access to SMAs and increase use of SMAs by providers. Although numerous studies have reported the benefits of SMAs to patients not all patients have the opportunity to participate. The burden of attending repeated SMAs could be alleviate by offering SMAs at more locations closer to where patients live and at a wider range of times.

## Conclusions

Even though numerous studies have highlighted the benefits of participation in SMAs to patients with chronic illnesses such as heart failure, diabetes and blood pressure [7–10, 19–27] there is room for further spread of this type of care delivery model. These findings illustrate the provider experience and further detailing the patient experience could inform facilitation and implementation strategies to encourage providers to lead and/or refer patients to SMAs. Further research is warranted to investigate the implementation of SMAs for HF at new sites.

### Practice implications

These findings highlight benefits for providers in participating and leading the HF SMAs and present some strategies for improving SMAs and the presentation of the benefits of SMAs to new clinical teams.

## Supporting information

**S1 Appendix. Interview guide.**
(DOCX)

**S2 Appendix. Agenda for debriefing post interview meetings.**
(DOCX)

## Acknowledgments

The data were presented at American Public Health Association (APHA) 2017 Annual Meeting. We acknowledge the study respondents for their time in completing the study. The views expressed in this manuscript are those of the authors and do not reflect the official policy of the U.S. Government and/or Department of Veterans Affairs.

## Author Contributions

**Conceptualization:** Lauren Stevenson, Julie Gee, Kimberley Schaub, Sharon LaForest, Tracey H. Taveira, Sandesh Dev, Amy Barrette, Karen Oliver, Wen-Chih Wu.

**Data curation:** Jeri Jewett-Tennant, Jeneen Shell-Boyd, Lauren Stevenson, Rene Hearns, Melanie Parent, Sherry L. Ball.

**Formal analysis:** Vanessa Marshall, Jeri Jewett-Tennant, Jeneen Shell-Boyd, Lauren Stevenson, Melanie Parent, Sandesh Dev, Sherry L. Ball.

**Funding acquisition:** Lisa Cohen, Sandesh Dev, Wen-Chih Wu.

**Investigation:** Vanessa Marshall, Jeri Jewett-Tennant, Jeneen Shell-Boyd, Lauren Stevenson, Tracey H. Taveira, Wen-Chih Wu, Sherry L. Ball.

**Methodology:** Vanessa Marshall, Jeri Jewett-Tennant, Lauren Stevenson, Julie Gee, Kimberley Schaub, Sharon LaForest, Lisa Cohen, Amy Barrette, Karen Oliver, Wen-Chih Wu, Sherry L. Ball.

**Project administration:** Rene Hearns, Julie Gee, Melanie Parent, Sherry L. Ball.

**Resources:** Kimberley Schaub, Sharon LaForest, Tracey H. Taveira, Lisa Cohen, Sandesh Dev, Amy Barrette, Karen Oliver, Wen-Chih Wu.

**Supervision:** Wen-Chih Wu, Sherry L. Ball.

**Writing – original draft:** Kimberley Schaub, Sharon LaForest, Tracey H. Taveira, Lisa Cohen, Sandesh Dev, Amy Barrette, Karen Oliver, Wen-Chih Wu, Sherry L. Ball.

**Writing – review & editing:** Vanessa Marshall, Jeri Jewett-Tennant, Jeneen Shell-Boyd, Lauren Stevenson, Rene Hearns, Julie Gee, Kimberley Schaub, Sharon LaForest, Tracey H. Taveira, Lisa Cohen, Melanie Parent, Sandesh Dev, Amy Barrette, Karen Oliver, Wen-Chih Wu, Sherry L. Ball.

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
