## [Decision Letter · Decision Letter 0]

7 May 2021

PONE-D-21-04750

Healthcare providers experiences with shared medical appointments for heart failure

PLOS ONE

Dear Dr. Ball,

Thank you for submitting your manuscript to PLOS ONE. After careful consideration, we feel that it has merit but does not fully meet PLOS ONE’s publication criteria as it currently stands. Therefore, we invite you to submit a revised version of the manuscript that addresses the points raised during the review process.

We look forward to receiving your revised manuscript.

Kind regards,

Tareq Mukattash

Academic Editor

PLOS ONE

Journal Requirements:

2)  We note that you have indicated that data from this study are available upon request. PLOS only allows data to be available upon request if there are legal or ethical restrictions on sharing data publicly. For information on unacceptable data access restrictions, please see http://journals.plos.org/plosone/s/data-availability#loc-unacceptable-data-access-restrictions.

3) We noted in your submission details that a portion of your manuscript may have been presented or published elsewhere. [The data were presented at American Public Health Association (APHA) 2017 Annual

291 Meeting.]

Reviewers' comments:

Reviewer's Responses to Questions

**Comments to the Author**

1. Is the manuscript technically sound, and do the data support the conclusions?

Reviewer #1: Partly

2. Has the statistical analysis been performed appropriately and rigorously? 

Reviewer #1: N/A

3. Have the authors made all data underlying the findings in their manuscript fully available?

Reviewer #1: No

4. Is the manuscript presented in an intelligible fashion and written in standard English?

Reviewer #1: No

5. Review Comments to the Author

Reviewer #1: April, 2021

Dear Editor,

Thank you for the invitation to review this manuscript which reports the results of a qualitative study exploring the Shared Medical Appointments (SMAs) experience from healthcare providers’ perspectives. The topic is interesting and of contemporary importance. However, the paper lacks important standards of rigorous scientific research. Kindly find my comments below:

1. Introduction:

- Line 57: suggest to define patient self-management in heart failure and its relationship to SMAs.

- Line 60: provide a reference at the end of statement and elaborate more on comprehensive patient care and how to deliver such care.

- Line 61: SMAs are the focus of the paper, yet I was not able to find a clear definition of the concept SMA. How often is SMA applied? At what disease states is it more helpful? Is it a novel concept or has it been long applied in clinical practice? Need to clarify the context where SMAs are most helpful to get the reader to better understand the concept especially if they are not familiar with this kind of shared patient care.

- Line 66: need to add a reference proving that active engagement in SMAs was associated with improved patient self-management.

- Line 81: was this really the focus of the paper? I didn’t appreciate any future direction for establishing new SAMs, yet most of the discussion was related to insights for improving already established SMAs. Need to refine aims to better align with the actual reported information.

2. Materials and Methods: overall, the methods section was poorly constructed and it failed to describe important information that are needed to replicate the study. I am listing major comments below:

- Line 83: the number of included sites must be listed at the very beginning of the section (i.e. under the setting part line 91), however, I wasn’t able to find it until line 106.

- Line 85: “patients were randomized to either SMA or usual care”. This is kind of vague; how often were the SMAs sessions held and how often were HF patients seen (in usual care) post hospital discharge? Did you conduct the SMA interviews with providers right after patients were seen, why did you report patient randomization? Need to clarify.

- Lines 95 and 96: patients were referred to SMAs by physicians, nurses, clinical RPhs, PAs, NPs while in line 105 dieticians and clinical psychologists were also listed; does this imply that those (i.e. dieticians and psychologists) were SMA leaders rather than SMAs referral points? Need to clarify.

- Lines 100-101: Convenience sampling while all provided names included? Does this imply that there were other providers’ names supplied but not included in SMA interviews? Or did you mean using convenience sampling to collect the providers’ names or sites? Need to clarify as this is confusing.

- Line 104: helpful to provide response rate (out of 33 invited, 24 accepted and were interviewed, ~ 73%).

- Line 108: Table 1 could have included more informative data related to sites (or even demographics of SMA leaders at different sites); years of providing SMA, hospital bed size, number of providers involved in SMAs, total number of HF seen throughout the study period vs those referred to SMA among others. This could have been even used in sub-analysis of facilitators and barriers as listed as aims of the present study in line 75.

- Line 110: were signed consents collected even from those who were interviewed over the phone? Need to clarify how written consents were obtained. Regarding consents, each participant was asked to provide two consents: one written and another audio taped as consent to record the interview, right? Clarify.

- Line 112: need to elaborate more on KAP framework, provide a reference and define context specific KAP.

- Line 113: clarify neutral domain.

- Line 114-116: how did you ensure consistency in interviewing process among three interviewers? Line 114 reported that 3 interviewers conducted the interviews while 116 noted that 2 trained interviewers conducted the interview; kind of confusing, were the three interviewers SMA interview leaders and note-takers at the same time or were they only leaders and you had 2 others to conduct the actual interview and take notes?

- Line 122: was this a condition for conducting the interview? Did you ensure lack of bias through assigning each interviewer to conduct the interview at a different site form interviewers’ affiliation? If not need to highlight.

- Line 128: team-based analysis; were those the same 3 interviewers together? Or a new team? Revise used terms, define as needed, and make sure to be consistent (team-based vs interviewers vs analysists?)

- Line 134: piloting? How was this done, this was not mentioned earlier? Was the study referenced in 7 (Cohen et al, 2017) the pilot study? Clarify as some information in the current paper and that published by Cohen are contradictory and lack accuracy (check my note related to S1 below for further clarifications).

- Line 146: Table 2 legend included the word “initial” which gives the impression that the listed domains are those of the primary template (i.e the 6 domains) while it actually listed the final domains retained after final matrix analysis excluding SMA attitude, please modify the legend accordingly.

3. Results:

- Lines 164-167: very long and hard to follow.

- Line: 203: confusing, need to revise.

4. Discussion: very concise and lack comprehensive comparison to other published studies; suggest to better align with the results of the study. Some of the results need to be discussed at length.

5. Conclusion: It is important to make sure that conclusions are grounded on the study findings.

6. S1: it is very important to clarify the association between the present study and that listed as reference 7; similarities and differences. When I tried to refer to the published paper by Cohen et al, 2017, most information were contradictory. I was not able to comprehend if that study was more of a pilot study to the current one or were they the same. Examples of contradictory information include: number of questions included in the S1, number of research assistants conducting the interviews, number of interviewed providers…..

I hope you find some of the provided insights useful in improving the scientific content of the paper. Besides, the paper needs extensive copy-editing to improve the flow of information.

Kind regards,

Yours sincerely,

Rawand A. Khasawneh

Assistant Professor of Clinical Pharmacy

Faculty of Pharmacy

Jordan University of Science and Technology

Irbid- Jordan

6. PLOS authors have the option to publish the peer review history of their article (what does this mean?). If published, this will include your full peer review and any attached files.

Reviewer #1: No

---

## [Author Response · Author response to Decision Letter 0]

9 Jul 2021

The following includes edits to our manuscript addressing each concerned raised by the reviewer: 

1. Introduction:

- Line 57: suggest to define patient self-management in heart failure and its relationship to SMAs.

This definition has been added.

- Line 60: provide a reference at the end of statement and elaborate more on comprehensive patient care and how to deliver such care.

Reference and explanation included.

- Line 61: SMAs are the focus of the paper, yet I was not able to find a clear definition of the concept SMA. How often is SMA applied? At what disease states is it more helpful? Is it a novel concept or has it been long applied in clinical practice? Need to clarify the context where SMAs are most helpful to get the reader to better understand the concept especially if they are not familiar with this kind of shared patient care.

The definition and studies involving are now introduced in more detail in paragraph 2,3 and 4 of the introduction. 

- Line 66: need to add a reference proving that active engagement in SMAs was associated with improved patient self-management.

Dickman et al., 2012 has been added as a reference showing self-reported improvements in management of a chronic illness. Specifically, participation in SMAs over four months increased exercise time and patient’s success at reaching measurable goals. 

- Line 81: was this really the focus of the paper? I didn’t appreciate any future direction for establishing new SAMs, yet most of the discussion was related to insights for improving already established SMAs. Need to refine aims to better align with the actual reported information.

The introduction has been edited to reflect the main goal of this paper which is to describe provders’ experience leading and referring patients to SMAs. 

2. Materials and Methods: overall, the methods section was poorly constructed and it failed to describe important information that are needed to replicate the study. I am listing major comments below:

- Line 83: the number of included sites must be listed at the very beginning of the section (i.e. under the setting part line 91), however, I wasn’t able to find it until line 106.

Added to the first line of the Materials and methods section. 

- Line 85: “patients were randomized to either SMA or usual care”. This is kind of vague; how often were the SMAs sessions held and how often were HF patients seen (in usual care) post hospital discharge? Did you conduct the SMA interviews with providers right after patients were seen, why did you report patient randomization? Need to clarify.

This information has been removed as it is not relevant to this qualitative work. 

- Lines 95 and 96: patients were referred to SMAs by physicians, nurses, clinical RPhs, PAs, NPs while in line 105 dieticians and clinical psychologists were also listed; does this imply that those (i.e. dieticians and psychologists) were SMA leaders rather than SMAs referral points? Need to clarify.

This has been clarified.

- Lines 100-101: Convenience sampling while all provided names included? Does this imply that there were other providers’ names supplied but not included in SMA interviews? Or did you mean using convenience sampling to collect the providers’ names or sites? Need to clarify as this is confusing.

- Line 104: helpful to provide response rate (out of 33 invited, 24 accepted and were interviewed, ~ 73%).

Reference to convenience sampling has been removed. All provider whose names were supplied were contacted. 

- Line 108: Table 1 could have included more informative data related to sites (or even demographics of SMA leaders at different sites); years of providing SMA, hospital bed size, number of providers involved in SMAs, total number of HF seen throughout the study period vs those referred to SMA among others. This could have been even used in sub-analysis of facilitators and barriers as listed as aims of the present study in line 75.

More information regarding each site was not obtained. 

- Line 110: were signed consents collected even from those who were interviewed over the phone? Need to clarify how written consents were obtained. Regarding consents, each participant was asked to provide two consents: one written and another audio taped as consent to record the interview, right? Clarify.

The process of collecting consents has been clarified. 

- Line 112: need to elaborate more on KAP framework, provide a reference and define context specific KAP.

An explanation of how the KAP framework was to organize the qualitative data has been included. 

- Line 113: clarify neutral domain.

Neutral domains are qualitative data categories without positive or negative attributes; this is now included in the manuscript. 

- Line 114-116: how did you ensure consistency in interviewing process among three interviewers? Line 114 reported that 3 interviewers conducted the interviews while 116 noted that 2 trained interviewers conducted the interview; kind of confusing, were the three interviewers SMA interview leaders and note-takers at the same time or were they only leaders and you had 2 others to conduct the actual interview and take notes?

More details have been added to clarify how interviewers and analysts worked together.

- Line 122: was this a condition for conducting the interview? Did you ensure lack of bias through assigning each interviewer to conduct the interview at a different site form interviewers’ affiliation? If not need to highlight.

The fact that interviewers and analysts had no prior relationship with respondents at any of the three sites is now included.

- Line 128: team-based analysis; were those the same 3 interviewers together? Or a new team? Revise used terms, define as needed, and make sure to be consistent (team-based vs interviewers vs analysists?)

More details have been added to clarify how interviewers and analysts worked together.

- Line 134: piloting? How was this done, this was not mentioned earlier? Was the study referenced in 7 (Cohen et al, 2017) the pilot study? Clarify as some information in the current paper and that published by Cohen are contradictory and lack accuracy (check my note related to S1 below for further clarifications).

The word piloting was misleading and has been removed. 

- Line 146: Table 2 legend included the word “initial” which gives the impression that the listed domains are those of the primary template (i.e the 6 domains) while it actually listed the final domains retained after final matrix analysis excluding SMA attitude, please modify the legend accordingly.

The legend has been modified. 

3. Results:

- Lines 164-167: very long and hard to follow.

- Line: 203: confusing, need to revise.

The results section has been edited extensively. 

4. Discussion: very concise and lack comprehensive comparison to other published studies; suggest to better align with the results of the study. Some of the results need to be discussed at length.

The discussion section has been expanded.

5. Conclusion: It is important to make sure that conclusions are grounded on the study findings.

The conclusions have been edited to align with the findings. 

6. S1: it is very important to clarify the association between the present study and that listed as reference 7; similarities and differences. When I tried to refer to the published paper by Cohen et al, 2017, most information were contradictory. I was not able to comprehend if that study was more of a pilot study to the current one or were they the same. Examples of contradictory information include: number of questions included in the S1, number of research assistants conducting the interviews, number of interviewed providers…..

The Cohen et al. 2017 is a completely separate study.

Besides, the paper needs extensive copy-editing to improve the flow of information.

Extensive edits have been made throughout the manuscript.

Thank you for your consideration of this revised manuscript.

Respectfully,

Sherry L Ball, PhD

---

## [Decision Letter · Decision Letter 1]

21 Jan 2022

Healthcare providers experiences with shared medical appointments for heart failure

PONE-D-21-04750R1

Dear Dr. Ball,

We’re pleased to inform you that your manuscript has been judged scientifically suitable for publication and will be formally accepted for publication once it meets all outstanding technical requirements.

Kind regards,

Tareq Mukattash

Academic Editor

PLOS ONE

Additional Editor Comments (optional):

Reviewers' comments:

Reviewer's Responses to Questions

**Comments to the Author**

1. If the authors have adequately addressed your comments raised in a previous round of review and you feel that this manuscript is now acceptable for publication, you may indicate that here to bypass the “Comments to the Author” section, enter your conflict of interest statement in the “Confidential to Editor” section, and submit your "Accept" recommendation.

Reviewer #1: All comments have been addressed

2. Is the manuscript technically sound, and do the data support the conclusions?

Reviewer #1: Yes

3. Has the statistical analysis been performed appropriately and rigorously? 

Reviewer #1: N/A

4. Have the authors made all data underlying the findings in their manuscript fully available?

Reviewer #1: No

5. Is the manuscript presented in an intelligible fashion and written in standard English?

Reviewer #1: Yes

6. Review Comments to the Author

Reviewer #1: I believe the authors did a good job addressing all points addressed from my side.

- The introduction was further expanded to include relevant suggested information.

- The methods section was greatly improved to make it possible to duplicate the study in the future.

- Conclusions are presented in appropriate way and are congruent with the study findings. The authors reconstructed this section based on the results as suggested by the reviewer.

- The authors copyedited the paper to improve the flow of information.

7. PLOS authors have the option to publish the peer review history of their article (what does this mean?). If published, this will include your full peer review and any attached files.

Reviewer #1: No

---

## [Editor Report · Acceptance letter]

28 Jan 2022

PONE-D-21-04750R1 

Healthcare providers experiences with shared medical appointments for heart failure 

Dear Dr. Ball:

I'm pleased to inform you that your manuscript has been deemed suitable for publication in PLOS ONE. Congratulations! Your manuscript is now with our production department. 

Kind regards, 

on behalf of

Dr. Tareq Mukattash 

Academic Editor

PLOS ONE